# Oxidative Stress and Aging as Risk Factors for Alzheimer’s Disease and Parkinson’s Disease: The Role of the Antioxidant Melatonin

**DOI:** 10.3390/ijms24033022

**Published:** 2023-02-03

**Authors:** Jana Tchekalarova, Rumiana Tzoneva

**Affiliations:** 1Institute of Neurobiology, Bulgarian Academy of Sciences, Acad. G. Bonchev Street, Block 23, 1113 Sofia, Bulgaria; 2Institute of Biophysics and Biomedical Engineering, Bulgarian Academy of Sciences, Acad. G. Bonchev Street, Block 21, 1113 Sofia, Bulgaria

**Keywords:** oxidative stress, aging, Alzheimer’s disease, Parkinson’s disease, melatonin

## Abstract

Aging and neurodegenerative diseases share common hallmarks, including mitochondrial dysfunction and protein aggregation. Moreover, one of the major issues of the demographic crisis today is related to the progressive rise in costs for care and maintenance of the standard living condition of aged patients with neurodegenerative diseases. There is a divergence in the etiology of neurodegenerative diseases. Still, a disturbed endogenous pro-oxidants/antioxidants balance is considered the crucial detrimental factor that makes the brain vulnerable to aging and progressive neurodegeneration. The present review focuses on the complex relationships between oxidative stress, autophagy, and the two of the most frequent neurodegenerative diseases associated with aging, Alzheimer’s disease (AD) and Parkinson’s disease (PD). Most of the available data support the hypothesis that a disturbed antioxidant defense system is a prerequisite for developing pathogenesis and clinical symptoms of ADs and PD. Furthermore, the release of the endogenous hormone melatonin from the pineal gland progressively diminishes with aging, and people’s susceptibility to these diseases increases with age. Elucidation of the underlying mechanisms involved in deleterious conditions predisposing to neurodegeneration in aging, including the diminished role of melatonin, is important for elaborating precise treatment strategies for the pathogenesis of AD and PD.

## 1. Introduction

### 1.1. Oxidative Stress, Alzheimer’s Disease (AD), and AD-like Pathogenesis in Aging

Alzheimer’s disease (AD) is one of the most closely related to aging brain neurodegenerative disease. Like other neurodegenerative illnesses, AD is characterized by progressive worsening of cognitive capacity and represents the most frequent condition of dementia [1]. The extracellular beta-amyloid protein (Aβ) deposition and intracellular accumulation of hyper-phosphorylated tau protein (p-Tau) are the common markers of AD that positively correlate with behavioral symptoms of progressive cognitive decline. 

The most popular hypothesis of underlying mechanisms of AD onset is the formation of Aβ plaque [2,3]. Different factors were suggested to contribute to Aβ deposition in aging, with enormous oxidative stress considered the most frequent factor for the appearance of AD symptoms [4,5,6]. Evidence revealing abnormally elevated markers of oxidative stress in conditions associated with AD supports the hypothesis that oxidative stress has a crucial impact on the pathogenesis of this disorder [7,8]. The extracellular accumulation of Aβ and impaired mitochondria function could trigger free radical formation [9]. However, it is still a matter of debate and speculation whether oxidative stress precedes the pathological accumulation of Aβ or vice versa. The neurotoxicity of the latter triggers excessive oxidative stress and consequent neuronal loss. In vitro and in vivo models and clinical data reviewed in this subsection suggest a bi-directional “vicious cycle-like” relationship between the two phenomena. An excellent review by Praticò (2008) [10] summarized the experimental and clinical evidence supporting the close relationship between excessive oxidative stress and AD markers such as Aβ and p-Tau triangle forms. From one side, oxidative stress might be considered a prerequisite of early AD symptoms associated with membrane lipid peroxidation, mitochondrial dysfunction, and cellular loss. On the other side, the extracellular deposition of Aβ and the accompanied formation of tau-triangles evokes neurotoxicity via enhancement of free radical production. The hypothesis about the “vicious cycle” maintained by interacting oxidative stress and accumulated AD pathological factors (Aβ and p-Tau) is supported by findings from cell cultures and animal models. 

Several mechanisms were suggested to contribute to the role of the disturbed endogenous antioxidant defense system in producing Aβ-deposit. There is an interrelation between neuro-inflammation and sustained oxidative stress in aging; one of the two phenomena can trigger the other and vice versa [11,12]. Thus, the oxidants-induced stimulation of proinflammatory genes, such as gene transcription for interleukin-1 (IL-1) in glial cells, could be a threshold factor for initiating Aβ accumulation [12,13]. 

In physiological conditions, astrocytes are responsible for maintaining the blood-brain barrier (BBB) and clearance of extracellular Aβ plaque [14]. However, under a high level of oxidative stress, the proper functioning of these glial cells is disturbed, leading to impaired BBB and Aβ neutralization [15]. Furthermore, the aging process is vulnerable to poor BBB function due to an attenuated antioxidant defense system and consequently damaged astrocytes [16] that could trigger abnormal extracellular Aβ deposition. Another mechanism underlying the oxidative stress-induced Aβ accumulation might be associated with the increased activity of two proteolytic enzymes that are crucial for plaque generation: β- and γ-secretase [6,10]. Lipid peroxidation of cell membranes triggers Aβ production by increasing BACE1 expression and γ-secretase activity. 

An abnormal level of oxidation products from macromolecules (proteins, DNA, RNA, lipid) was demonstrated in close relationship with Aβ1–40 and Aβ1–42 in the brain regions that are most vulnerable to AD: the hippocampus and the cortex [6,8]. The most common markers of oxidation in AD were carbonylated membrane proteins and protein and lipoic acid oxidation due to a reaction with 4-hydroxynonenal (4-HNE), a lipid peroxidation product [17]. Mitohondrial dysfunction in AD was associated with the oxidation of DNA, including elevated free radical compounds (i.e., 8-oxo-2-dehydroguanine, 8-hydroxyadenine, 5-hydroxyuracil) in the most vulnerable cortical regions [18].

### 1.2. The Role of Enormous Oxidative Stress in Triggering Aβ Accumulation and p-Tau (In Vitro and In Vivo Models)

Most of the findings in the literature support the presumption that increased oxidative stress is the factor triggering the Aβ deposition (Table 1). The impairment of cell membranes due to the oxidation of neuronal membrane proteins and lipid peroxidation, as well as oxidation of the low-density lipoprotein receptor-related protein, could initiate AD pathogenesis associated with suppression of Aβ clearance, Aβ aggregation, and formation of extracellular plaque. The oxidative stress-induced changes of APP metabolism directed to pathological accumulation of Aβ via abnormal release of sAPPα and sAPPβ were demonstrated in cell cultures from transgenic Tg2576 mouse cerebral endothelial cells [19] (Table 1). The results suggested the involvement of the p42/44 MAPK phosphorylation in the mechanism underlying the Aβ deposition. Recently, Ma et al. (2022) [20] found that elevated oxidative stress, detected near Aβ deposit, predisposed to neurotoxicity and neuronal loss in neurons observed in real-time from mouse brains of the AD model. Liang et al. (2005) [21] results suggest that PGE2 EP2 receptor-related signaling is responsible for oxidative stress, resulting in BACE1 processing and formation of Aβ plaques in aged but not young male and female mice used as a model of familial AD.

Lecanu (2006) [22] used an intricate in vivo approach of the disturbed endogenous antioxidant system to demonstrate that increased oxidative stress is a prerequisite to initiating the accumulation of fibrillar Aβ1–42 and consequent neurotoxic events that model AD. Thus, although a long-term intracerebroventricular (icv) infusion of Aβ1–42 or pro-oxidative cation Fe2+ + Aβ1–42 in young adult rats was ineffective in triggering the production of AD markers in the brain, a combination with the inhibitor of glutathione (GSH) synthesis led to a memory decline and accompanied neurotoxicity-related changes (neuronal loss, increased level of p-Tau and Aβ accumulation). This data confirms the hypothesis that disturbed homeostasis of the defense antioxidant system triggers the formation of Aβ plaque and further neurotoxicity associated with neuronal damage and memory impairment. 

The work of Resende et al. (2008) [23] with the triple-transgenic mouse model supports the view that enormous oxidative stress, associated with a destroyed antioxidant defense system, appears before Aβ plaques and the development of AD symptoms. The authors reported that a disturbed homeostasis related to pro-oxidants/antioxidant balance was evident before forming Aβ plaques and p-Tau. The transgenic mice had an attenuated level of endogenous antioxidants GSH and vitamin E, concomitant with an increased membrane peroxidation and activity of the glutathione peroxidase (GPx) and superoxide dismutase (SOD). The latter process might represent an adaptive mechanism that was also reported by Pappolla et al. (1998) [24] in another transgenic mouse model of AD.

Walten et al. (2007) [25] elaborated a model of AD in rats chronically treated with aluminum from 12 months to 22 months (Table 1). The author found that the critical period for detecting high oxidative burden in the hippocampal neurons of rats with memory decline was in the early stages of aluminum accumulation (Stage I and Stage III). In contrast, no changes were observed at late Stage 5 of aluminum load, which was characterized by irreversible morphological aberration and neuronal death. In this work, an HNE marker was used for immunohistochemical detection of oxidative stress-induced cellular damage, which is known to be closely associated with the appearance of Aβ plaques and the formation of neurofibrillary tangles [32,33]. The aluminum-induced enormous oxidative stress in aged rats was accompanied by the formation of p-Tau [25]. However, the aluminum-induced model in old rats with memory impairments was not accompanied by an appearance of Aβ toxic aggregates that species-related structural differences with human Aβ sequence might explain.

Nevertheless, aluminum-evoked histochemical changes in the aged rat hippocampus resemble the early pathogenesis preceding the formation of Aβ deposit and p-Tau-related tangles in the human brain. The critical role of aluminum load for the onset of the earliest pathological changes predicting the further occurrence of AD-like symptoms was also demonstrated in transgenic mice (Tg2576) treated with aluminum in food [26]. The AD-like pathogenesis with the administration of this metal in the diet of Tg2576 was more severe, with oxidative damage in the brain preceding the formation of senile plaques, confirming the neurotoxic effect of aluminum exerted through a mechanism involving oxidative stress.

Recently, we elaborated another approach to attenuate the endogenous antioxidant defense system related to melatonin by removing the pineal gland in adult Sprague Dawley rats, concomitant to icv infusion of Aβ1–42 [34]. The impaired antioxidant system was confirmed in our model of melatonin deficit + icvAβ1–42 and resulted in more severe behavioral changes, Aβ deposition, and disturbed oxidant/antioxidant balance in the hippocampus. Our findings suggest that melatonin deficit, associated with elevated oxidative stress, is a predisposing factor reinforcing the Aβ1–42 plaque formation.

### 1.3. Clinical Data Supporting the Role of Enormous Oxidative Stress for Induction of Aβ Accumulation and p-Tau

Numerous reports suggest a close link between detected Aβ1–40 and Aβ1–42 deposits and elevated levels of oxidized proteins, lipids, and nucleic acids in the hippocampus and cortex of patients with AD [35,36,37,38,39,40,41]. Literature data in subjects with mild cognitive impairment (MCI) or patients with AD at an early stage support the hypothesis that the enormous oxidative stress might be a triggering factor for the progression of AD and, thereby, might represent a valuable marker for detection of early symptoms of the diseases. Thus, Abe et al. (2002) [42] reported that there is a positive correlation between the level of oxidized RNA in cerebrospinal fluid and the severity of pathology in newly diagnosed patients with AD (Table 2). Moreover, Ding et al. (2005) [38] found that the oxidized RNA in ribosomes was located preferably in brain regions associated with cognitive functions in patients in the early stage of AD. DNA oxidation was also demonstrated in the leukocytes of subjects with MCI and patients in the early stage of AD [43]. Lipid peroxidation, detected in body fluids (urine, plasma, and CSF), was higher in subjects with MCI compared to healthy aged subjects [44]. Oxidized proteins and lipids were found in the superior and middle temporal gyri of patients with MCI and early stage of AD [39]. Furthermore, they were also detected as vulnerable to the accumulation of Aβ regions early in the AD onset [45]. Disturbed mitochondria function could trigger AD-related pathological events by initiating free radical species in this organelle [46,47]. Moreover, lipid peroxidation was more frequent in senile plaques, wheras protein oxidation was positively correlated with the severity of memory decline. Subjects with MCI and patients with AD had astrocytes with massive oxidative levels [48] and damaged antioxidant defense systems in plasma and red blood cells [49,50].

Misonou et al. (2000) [40] supported the hypothesis for oxidative stress-induced progression of Aβ intracellular deposition by using human neuroblastoma SH-SY5Y cells (Table 2). Further, Nunomura et al. (2000) [41] demonstrated the role of oxidative stress for the Aβ plaque accumulation related to Down syndrome pathology. These authors detected increased oxidative stress before the Aβ1–42 deposit, whereas Aβ plaque accumulation was accompanied by a reduced but not enhanced stress-induced impairment. However, the coincidence of intracellular neuronal origin of both Aβ1–42 and oxidized nucleosides and protein [52,53] suggests the critical role of Aβ proteins within the neurons for oxidative stress-associated damage. Furthermore, neurons received from brain structures with Aβ plaque were characterized by enormous oxidative stress [6], whereas the cerebellum showed less Aβ and low levels of markers of oxidative stress [54,55,56]. 

### 1.4. The Antioxidant Role of Melatonin in the Pathogenesis of AD and Aging (In Vitro and In Vivo Experimental Data and Data in Humans) 

The scavenger role exerted by non-enzymatic antioxidant compounds is essential for homeostatic regulation and elimination of pro-oxidant molecules. The reduced capacity of the antioxidant scavenger mechanism brings about the elevation of free radical levels that could trigger DNA damage and neuronal loss. The role of melatonin as a powerful antioxidant is well-known and described in many reviews [57,58,59]. Melatonin can exert its antioxidant activity either via direct scavenger action or indirectly via affecting pro- and endogenous antioxidant markers in cells [60,61]. Additonally, the hormone can stimulate gene expression of GPx and glutathione reductase (GRd) [62].

Melatonin exerted antioxidant activity in cell cultures and decreased lipid peroxidation [62,63]. Compared to other antioxidants, such as vitamin E, that could be auto-oxidized, melatonin and other endogenous indole compounds (indole-3-propionic acid) have the advantage of directly neutralizing hydroxyl radicals, which are the intracellular signals in Aβ-related neurotoxicity, and thereby can behave as an endogenous electron donor [64]. 

Cell viability was used as a reliable indicator of the protective capacity of antioxidants, including melatonin, against Aβ-triggered toxicity because there is a close relationship between the severity of the cognitive decline in AD and the rate of neuronal loss [65]. The antioxidant action and cellular protection of melatonin caused by Aβ toxicity were demonstrated in cell cultures from human platelets [62], murine N2a neuroblastoma cells [66], as well as oxidative stress-induced impairment of mitochondrial DNA [66], and DNA fragmentation in C6 cells [63] (Table 3). The antioxidant properties of this hormone promote the protection of astroglial-like cells, rat primary hippocampal neurons, human neuroblastoma SK-N-SH cells, and PC 12 rat pheochromocytoma cells [63] against Aβ-related neurotoxicity. Pappolla et al. (1997) [66] used nuclear magnetic resonance spectroscopy to explain the underlying mechanism of melatonin-related prevention of Aβ fibrillation through impaired imidazole-carboxylate links in Aβ protein. Furthermore, melatonin (MT) receptors were reported not to be involved in the neuroprotective mechanism of action of the hormone in primary hippocampal neurons because of the lack of direct antioxidant effect of melatonin agonists associated with Aβ or other oxidative stress protocol [67]. However, Pappolla et al. [68] (2002) did not explore the possible participation of intracellular receptors in MT antioxidant function. The results of Pappolla et al. (1998) [24] agree with other in vitro data revealing the link between Aβ and enhanced oxidative stress, including membrane lipid peroxidation [69], mitochondrial dyshomeostasis [70], and reduced antioxidant enzyme activity [69,71]. Exposure of cultured neuroblastoma cells to melatonin had a powerful neuroprotective effect through attenuation of Aβ-induced cytotoxicity associated with cellular oxidative impairment and intracellular Ca^2+^ elevation [66] (Table 3). 

Five days after a procedure of icv infusion of protofibril Aβ, disturbed homeostasis of the antioxidant system was observed in the neocortex and the hippocampus of male Swiss mice [72] (Table 3). The authors used Th-T fluorescence to verify the close link between Aβ accumulation and increased oxidative stress and the lack of such interrelations between Aβ monomer, Aβ scrambled peptide, and oxidative stress markers. They hypothesized that the scavenger effect of the hormone on protofibrils leads to reduced free radical formations and protection of neurons. Therefore, melatonin’s antioxidant and neuroprotective potency was indirect in this case.

**Table 3 ijms-24-03022-t003:** The antioxidant role of melatonin in the pathogenesis of AD and aging (animal data and clinical).

In Vitro and In Vivo Models	Melatonin Treatment	Oxidative Stress	Beta-Amyloid	p-Tau Protein	References
human platelets membranes	Melatonin (1 and 2 mM)	lipid peroxidation ↓	Aβ + aluminum treatment	-	[62]
C6 astrocyte-like cell line	Melatonin (10^−5^ to 10^−7^ M)	lipid peroxidation↓	melatonin mitigated Aβ-related toxicity	-	[63]
murine N2a neuroblastoma cells	Melatonin (10 and 50 mM)	MDA ↓	-	-	[66]
murine N2a neuroblastoma cells	Melatonin (10 mM)	oxidative damage of mtDNA ↓	-	-	[67]
rat primary hippocampal neurons, human neuroblastoma SK-N-SH cells, and PC 12 rat pheochromocytoma cells	Melatonin (100 mM)	-	Aβ-associated cell viability	-	[68]
Neuro2A cell line	Melatonin (10, 50 mM)	melatonin—intracellular ROS ↓ induced by Nrf2	-	Protection of melatonin (10 and 20 µM) of p-Tau-exposed neurons; Melatonin suppresses GSK3b-mediated p-Tau	[73]
Transgenic mouse model of ADs before Aβ deposit	melatonin (10 mg/kg for four months)	thiobarbituric acid reactive substances, SOD and GSH↓ in brain cortex	melatonin—↓ Aβ-related neurotoxicity in brain cortex	-	[74]
Tg2576 8, 9.5, 11, and 15.5 month-month old mice	Melatonin (50 mg/kg) starting from 4-month-old mice	Nitration of proteins in 8-month-old mice↓	Aβ in 9.5, 11, and 15.5 month-month old mice↓	-	[75]
Tg2576 14-month-old mice	melatonin (16 µg/mL) for four months	8,12-isoprostane F2a-VI (lipid peroxidation product) not changed	Aβ—not change	-	[76]
Tg2576 mice	long-term with melatonin and aluminum	↑oxidative stress (not affected)	↑ Aβ deposit (not affected)	-	[77]
male Wistar rats, 3–4 months old with icv streptozotocin (3 mg/kg for 30 days)	Melatonin (10 mg/kg, i.p.)	-	Aβ deposit ↓ in the hippocampus	-	[78]
Sprague-Dawley rats	Melatonin 0.15 mg/kg, i.p. 40 days	-	Aβ deposit ↓	-	[79]
Wistar rats	Melatonin 10 and 20 mg/kg, once per week for four week	-	APP deposition ↓ brain (20 mg/kg melatonin)	-	[80]
Sprague-Dawley rats	Melatonin 30 mg/kg for 13 days	-	Protection against Aβ1–42-induced microvascular changes via receptors VEGFR1 and VEGFR2	-	[81]
Male Wistar rats	Melatonin (20 mg/kg/daily in drinking water for several days	Nitrite levels and lipid peroxidation products in hippocampal and cortical tissues↓	Suppresses Aβ-induced increased Il-6 and TNF-a levels in the brain	-	[82]
Sprague-Dawley 10-months rats with hippocampal Aβ_25–35_ (2 g/L) injections	Melatonin (0.1, 1, 10 mg/kg, i.g., ten days)	lipid peroxidation↓; SOD↑; GSHpx and GSH↑ in cytoplasm and mitochondria	-	-	[83]
C57BL/6 mice seven months-old treated with D-Galactose (DG) (s.c., 120 mg DG/mouse/daily for 49 days)	Melatonin (10 mg/kg from day one till day 88 after DG	GSH/GSSG ratio and SOD↑ and lipid peroxidation↓	Aβ plaques in the hippocampus ↓	-	[84]
PP/PS1 transgenic mice 6–9 months old	Melatonin (0.5 mg in drinking water) between 6th and 9th month	ROS↓ in brain	-	-	[51]

Age-related suppression of melatonin release from the pineal gland was associated with diminished adrenergic receptors in pinealocytes [59]. Aging has altered function of the primary circadian pacemaker in the mammalian, the suprachiasmatic nucleus, leading to a decreased control of melatonin synthesis from the pineal gland and disturbed sleep-wake synchronization [71,85]. Similar changes are also evident in AD, suggesting that aging represents a crucial factor in the development of this disease [85,86]. 

Melatonin supplementation was reported to reduce the production of p-Tau and p-Tau-associated neurotoxicity in the neuro2A cell line [73]. The antioxidant effect of melatonin via nuclear translocation of neuronal Nrf2, as well as its anti-inflammatory action in microglial cell line demonstrated by Das et al. (2020) [73], might be suggested an important neuroprotective mechanism of the hormone action against p-Tau neurotoxicity. 

The beneficial effect of melatonin supplementation on Aβ neurotoxicity in rat models of AD was reported in many studies. Feng et al. (2006) [74] reported a beneficial effect of melatonin treatment (10 mg/kg) in transgenic mice before the formation of Aβ plaque. Long-term treatment with this hormone for four months had potent antioxidant activity in the brain cortex via suppression of lipid peroxidation, SOD, and GSH. Thus, reduced oxidative stress was concomitant to reduced neuronal apoptosis, suggesting that suppression of oxidative stress before the development of AD-related neurotoxicity is an important target to prevent the pathogenesis of ADs. Notably, Matsubara et al. (2003) [75] reported that long-term treatment with this hormone in transgenic mice prevented the Aβ deposit in the cortex and the hippocampus. The authors used a protocol of continuous treatment with melatonin, used in pharmacological doses, starting from 4 months-old rats to different age periods i.e., 8-, 9.5-, 11-, and 15.5-month-old rats, respectively. As it was expected, a tendency for the elevation of Aβ levels in the aged brain was demonstrated, which was significantly suppressed by melatonin from the age of 9.5 months, when Aβ aggregation was detected. However, the excessive nitration of proteins was reduced by melatonin in the youngest group of 8 months but not in older groups, suggesting the lack of positive correlation between Aβ and protein nitration in the effect of melatonin, which is possibly related to an age-increased amyloid load that melatonin failed to prevent at the dose used in this protocol.

In contrast, Quinn et al. (2005) [76] also administered melatonin for four months but started the procedure at 14 months when the transgenic mice had already formed Aβ plaque. The author used a reasonable human dose of melatonin, which should not be higher than accepted in the clinic, with a peaked plasma level of 1000 pg/mL after a 3-mg amount. Quinn et al. (2005) [76] findings on Tg2576 mice indicate that the age period for melatonin supplementation is crucial for its beneficial effect. 

Further, Di Paolo et al. (2017) [77] confirmed the results of Quinn et al. (2005) [76], revealing that long-term oral administration of melatonin, concomitant with aluminum, on Tg2576 mice failed to prevent Aβ neurotoxicity and a high level of oxidative stress. These works raise the question that, although the prophylactic treatment with melatonin might be effective in preventing the propagation of AD pathogenesis via suppression of free radicals in specific brain regions, supplementing the hormone in older mice is ineffective in neutralizing Aβ deposition and excessive oxidative stress. 

Melatonin supplementation, applied for a short- or long-term period in different doses, reduced the accumulation of Aβ in the hippocampus in different rat strains [78,79,80] and attenuated the Aβ-induced impaired microvessels in the cerebral cortex and hippocampus and neurotoxicity [81], as well as Aβ-related neuroinflammation [82] and Aβ accumulation and associated memory decline [76]. However, few data on this hormone’s effects on AD-related oxidative stress were demonstrated [79,82,83]. The potent dose-dependent antioxidant activity of a subchronic melatonin administration in middle-aged rats with the hippocampal injection of Aβ25–35 was accompanied by a cognitive improvement in the Morris water test [83]. The protection exerted by melatonin against oxidative stress was also reported to be associated with the correction of cognitive decline in middle-aged C57BL/6 mice administered with D-galactose (DG) that modeled aging and AD [84]. Thus, melatonin inhibited Aβ deposit in the hippocampus and, related to this process, elevated oxidative stress in the DG model. Moreover, the hormone-attenuated acetylcholinesterase (AChE) activity might explain its beneficial effect on learning and memory, shown in an active avoidance test and water-maze test [84]. The treatment protocol (melatonin, 10 mg/kg, from day one till day 88 after DG) suggests that the antioxidant effect of the hormone was indirectly due to its prevention of Aβ accumulation in brain tissue and, thereby, neutralization of Aβ neurotoxicity associated with excessive production of free radicals. 

Rosales-Corral et al. (2003) [82] reported that other powerful antioxidants, such as vitamin C and vitamin E, exhibit comparable melatonin protection against intra-hippocampal infused fibrillar Aβ in young adult Wistar rats; the hormone had a more powerful anti-inflammatory response. The authors suggest that the latter effect of melatonin might not be mediated by MT receptors, as was reported earlier by Pappolla et al. (2002) [68] in cell cultures, but by intracellular activity [87]. 

Recently, Fan et al. (2022) [51] demonstrated that a major transcription factor involved in the process of autophagy/mitophagy TFEB was diminished in brain of the PP/PS1 transgenic mice while melatonin affected mitophagy via facilitated TFEB nuclear translocation. Mitophagy is a type of autophagy affecting impaired mitochondria degraded by lysosomes [88]. Mitophagy is a part of pathogenesis associated with both aging and AD [89]. There is a close relationship between the impaired mechanism of lysosome-associated autophagy and changed Aβ metabolism and Tau protein expression in AD [90]. The TFEB-induced elevation of ALP suppressed AD-related pathogenesis by mitigating Aβ deposition and p-Tau, memory decline, and enhancing enhance autophagy/mitophagy. 

Recently, we reported that melatonin supplementation corrected the pathogenesis of the pinealetomy + Aβ1–42 rat model, suggesting the crucial role of the melatonin system in the mitigation of oxidative stress and associated Aβ1–42 accumulation [34]. The hypothesis that melatonin deficit might predispose to the pathology of AD is supported by several preclinical and clinical reports [91]. The preclinical AD subjects and patients with the disease had decreased plasma melatonin levels due to either the disturbed turning of 5-HT to melatonin or the changed adrenergic receptor function. 

## 2. Parkinson’s Disease (PD)—Oxidative Stress, Pathogenesis

Parkinson’s disease (PD) is another age-related disease that is mainly caused by the selective loss of the dopaminergic (DA) pathway to the substantia nigra pars compacta (SNpc). The loss of DA leads to deregulation in the basal ganglia circuitries and to the appearance of motor symptoms such as bradykinesia and rigidity, resting tremor, and postural instability. PD is also characterized by the appearance of non-related motor functions and behavioral responses such as sleep disturbances, depression, and cognitive deficits [92]. The exact etiology of PD and the mechanisms that cause this disease remain not yet fully established [93]. There are two types of PD: familial (mutations in a number of genes) and sporadic [94]. Nevertheless, mitochondrial dysfunction, neuroinflammation, and environmental factors are strongly appreciated as key determinants of both familial and sporadic forms of the disease [95]. In both cases, oxidative stress, which is responsible for the increased production of ROS and reactive nitrogen species (RNS), is believed to be the common mechanism leading to cellular dysfunction as DA metabolism [96] or to iron accumulated in the SNpc [97] and eventual cell death in PD. 

For instance, dopaminergic neurons involved in catecholamine synthesis may cause the accumulation of excess free radicals and oxidative stress, thus leading to mitochondria dysfunction. Thus, dopamine oxidation and mitochondrial dysfunction are pathologically linked through oxidative stress and, together with defects in the clearance of abnormal protein aggregates (such as α-synuclein aggregation, which causes α-syn-rich Lewy bodies) [98]), are important factors for PD pathogenesis [99]. In recent years, a growing body of evidence has emerged for the crosstalk between oxidative stress and autophagy which is involved in PD pathogenesis [100]. Members of the microphthalmia-associated transcription factor family (MiT), including microphthalmia-associated transcription factor (MITF), transcription factor E3 (TFE3), and transcription factor EB (TFEB), play a central role in regulating cellular homeostasis in response to metabolic pressure and are considered master regulators of lysosomal signaling, which includes the mTOR pathways [101]. The members of MiT family interact between mitochondria and lysosome functions and therefore represent attractive targets for therapeutic approaches against PD. Recently, it was reported that a mild increase of ROS levels could activate mucolipin 1 (MCOLN1), a key calcium-conducting channel on the lysosome membrane, to initiate calcineurin-dependent activation of TFEB [102], which is identified as a master regulator of the ALP [103]. In turn, the TFEB-mediated induction of autophagy promotes the clearance of damaged mitochondria and the removal of excess ROS [102]. However, excessive ROS levels may cause lysosomal dysfunction and autophagic failure and lead to cell death [102]. Activation of TFEB has been shown to alleviate neurodegeneration in several in vitro and in vivo models of neurodegenerative diseases through lysosomal function enhancement and autophagy induction [104]. Thus, TFEB enhancers, such as Torin 1, a potent mammalian target of rapamycin (mTOR) inhibitor 29, and C1, a novel curcumin analog could be used as pharmacological approaches to treat PD [105].

Besides the genetic mutations that cause the increased oxidative stress described above, there are other factors that play a role in changing the physiological redox status and are recognized as risk factors for the development of PD. Among them is aging, which is responsible for accumulating a critical level of misfolded pathogenic proteins, which can cause neuronal damage associated with age-related impairment of mitochondrial function and subsequent increased production of ROS [106].

Epidemiological studies confirm the association between exposure to multiple pesticides commonly used in agriculture and increased risk of developing PD [99]. Although these toxins act through different mechanisms, they share a common characteristic of increased oxidative stress due to the increased production of ROS.

### 2.1. Animal Models of PD

Parkinson’s disease is a heterogenous disease varying in age, symptoms, and rate of progression. Studying the different aspects of this heterogenous disease requires to use of a variety of animal models. Animal models of PD contribute to the elucidation of the molecular pathways of neuronal degeneration and the unknown mechanisms by which oxidative stress contributes to the development of PD [107,108].

#### 2.1.1. Species-Specific Characteristics

There are basically three groups of animals that are used to model PD, including non-mammalian species, rodents, and non-human primates (NHPs). Each group has its advantages and limitations.

Non-mammalian models, such as drosophila and Caenorhabditis (C.) elegans, have also been used in some experimental PD studies because of their rapid reproductive cycle, easily generated genetic manipulations, well-manifested neuropathology, and behavior and due to low costs of maintenance [108]. Because of the difficulty of translating research conducted in non-mammalian species to humans, it is rarely used.

The majority of published PD animal studies involve rodents, as they are convenient to rear in laboratory conditions and have developed robust experimental protocols for working with different forms of drug preparations, generation of transgenic strains, and behavioral tests for measurement of motor activity, akinesia, and bradykinesia [109,110].

The non-motor symptoms in rodents can be monitored to assess sleep disturbance and weight loss [97]. To track the depressive-like behavior that modeled neuropsychiatric symptom a large set of behavioral tests was used [111]. 

Research involving NHPs provides valuable insight into the pathology of PD due to their anatomical and genetic similarity to humans [112]. Compared to rodents, NHPs have some advantages as the larger size and the longer life span, but at the same time, they require greater care and high costs and involve more complex ethical considerations. NHP models resemble neurotoxic or virus-mediated PD pathology that induces PD symptoms and behaviors similar to those in humans. 

Two approaches are used to model PD experimental animals: neuro-toxins and genetics. Neurotoxin-induced dopaminergic neurodegeneration models epigenetic factors that have been implicated in PD. This approach generally leads to a profound neuronal death in the SNpc that has a positive correlation with the deficit in motor and behavioral functions but lacks the formation of Lewy bodies [113]. In contrast, genetic-based models are characterized by divergence in neuronal damage, motor symptoms, and concomitant α-synuclein pathology. Genetic mutations or changes in gene expression can be modeled using transgenic animals or induced by viral transfection. Both neurotoxins and genetic approaches are applied in different animal species to model PD.

#### 2.1.2. Neurotoxic Models 

These PD models are induced by local or systemic administration of neurotoxins and may establish a link between oxidative stress and degeneration of dopaminergic neurons. 

MPTP (1-methyl-4-phenyl-1,2,3,6-tetrahydropyridine) is one of the most commonly used neurotoxins to model PD, which can be administered acutely or chronically [114]. Due to its lipophilic nature, it easily crosses the BBB and is taken up by glial cells. There it is converted into the toxic agent 1-methyl-4-phenylpyridinium (MPP+) by spontaneous oxidation [115]. It is transported and accumulated into dopaminergic cells, which can negatively affect the function of the mitochondrial complex I related to ATP production [116]. In this way, this neurotoxin can induce excessive oxidative stress and concomitant neuronal damage and inflammation. Therefore, this model is helpful for the assessment of disturbed mitochondrial function, homeostasis of oxidative stress, and inflammation closely related to the affected by the MPTP dopaminergic pathway. Although this neurotoxin MPTP is reliable in modeling symptoms of PD in mice and NHP, rats are resistant to moderate doses of MPTP, whereas higher doses increase mortality [115]. The acute dose of MPTP, administered intraperitoneally, causes loss of DAergic neurons, mainly in the SNpc [113]. Neurodegeneration occurs within hours and stabilizes within seven days. Thus, MPTP-induced dopaminergic degeneration in mice correlates with motor deficits, but these can be recovered within a few days after the acute dose, which appears to be a limitation on the duration of behavioral tests [110]. In addition, the MPTP mice model has also been used to study gut microbiota dysbiosis in PD. Transplantation of fecal microbiota from MPTP-treated mice into non-wild-type mice reduced striatal dopamine levels and induced motor impairments in the latter. In contrast, transplants of normal microbiota alleviated PD-like symptoms in MPTP-treated mice [117].

6-OHDA (6-hydroxydopamine) is an analogue of dopamine and norepinephrine [118]. It is important that 6-OHDA does not cross the BBB and must be administered directly into the brain to induce neurodegeneration. It selectively targets dopaminergic neurons [118]. The 6-OHDA induces oxidative stress by producing reactive oxygen species (ROS), such as superoxide radicals, hydroxyl radicals, and hydrogen peroxide, and causes cell death by inhibiting mitochondrial complex I [119]. Notably, this reaction is cata-lyzed by iron, and therefore the toxicity can be prevented by iron-chelating agents, such as vitamin E or MAO-B inhibitors (selegiline) [116].

The most recent and most debated neurotoxins used for modeling PD are pesticides and herbicides [120]. Emphasis has been made on rotenone, paraquat, and maneb as possible environmental causes of PD. 

Rotenone and paraquat are thought to cause DAergic degeneration by inducing oxidative stress. Rotenone probably acts via a complex I, while paraquat exerts its toxicity through cellular redox cycling [121]. Despite unclear mechanisms of action and how they cross BBB [122,123], the rotenone and paraquat models are used to induce parkinsonian pathology and to study the inhibition of inflammatory and oxidative stress pathways in adult rodents. 

Genetics plays an important role in PD pathogenesis, since disease-causing mutations have been identified through linkage analyses in familial PD. In contrast, genetic risk factors for idiopathic PD have been identified through association analyses in patients and controls [124]. A breakthrough in PD research was the discovery of the first gene (SNCA) associated with familial PD [125,126]. It encodes the protein α-synuclein, which is aggregated in Lewy bodies, and that is the way these animal models are called “α-synuclein models.” Since then, the identification of additional monogenic PD mutations has been used to study the effects of mutated proteins and to develop new reliable animal models. Since most of the genetic models are only effective at reproducing some of the PD distinguishing marks, the development of new genetic tools as viral vectors that can be used to introduce wild-type or mutated genes for targeted expression of a disease-associated protein allows for the generation of new genetic-based models. 

Overexpression α-synuclein can be induced by viral vectors. Among the advantages of this method are the induction of pathology in adulthood, targeting of the nigro-striatal system, and the possibility for adjustable dos-age [127]. Viral vectors such as a recombinant adeno-associated virus (rAAV) and lentivirus (LV)-based vectors have been used to transfer SNCA in rodents. Viral vector-mediated α-synuclein models have also been applied to mice, rats, and NHP (marmosets and macaques). In NHP, they demonstrate robust motor deficits, and they can become helpful preclinical models. In addition, the viral models have the advantage of producing α-synuclein pathology and can lead to the development of treatments targeting α-synuclein toxicity. A disadvantage of viral vector-based models is the unfavorable interaction with subsequent viral transductions used in gene therapy since exposure to the first viral vector can alter the response to future exposure to viral vectors by altering transfection and the reliability of experimental results.

### 2.2. The Antioxidant Role of Melatonin in the Pathogenesis of PD (Animal Data and In Vitro Studies) 

Oxidative stress is involved in neurodegeneration in PD through the activation of c-Jun N-terminal kinase (JNK) and c-Jun, which play an important role in cell death [128,129]. Attempts to use compounds with antioxidant properties in PD have included tocopherol (vitamin E), vitamin C, Coenzyme Q10, docosahexaenoic acid (DHA), Ginkgo biloba, or polyphenols found in green tea [130,131,132,133,134]. None of these have yielded convincing evidence for neuroprotective efficacy. In this respect, melatonin, with its direct (free radical scavenger) and indirect antioxidant properties (stimulating the synthesis of antioxidant enzymes and inhibiting that of pro-oxidant enzymes), shows more effectiveness than vitamins C and E [135]. It has been found that the melatonin levels in PD patients were lower than in healthy volunteers, and these decreased levels correlated with the severity of the disease [136] and its administration has been reported to improve sleep behavior [137] and non-motor disturbances in PD patients [138]. In animal PD models, melatonin is assumed to have neuroprotective activity toward DAergic neurons [139,140,141]. Several studies reported the effects of melatonin on oxidative stress in PD models, which are listed in Table 4.

Administration of melatonin (10 mg/ kg/day, i.p.) decreased MDA and DA neuron death and increased SOD, CAT, and GPx in the SN of 6-OHDA—rat model of PD [142]. In a similar study, melatonin was administrated at a dose of 10 mg/kg before and after the induction of the 6-OHDA—rat model [143]. The more pronounced anti-oxidative and anti-inflammatory effect of melatonin was found in the pre-treated group. In this way, the protective/preventive effect of melatonin on neurons was detected. The administrated melatonin after establishing the PD model was less effective. In confirmation of these results are the studies of Patki and Lau on the MPTP animal model [144] (2011), who reported that when the chronic Parkinsonian mice were pre-treated and continuously treated with melatonin (5 mg/kg/day, i.p.) for 18 weeks, the defects of mitochondrial respiration, ATP, and antioxidant enzyme levels detected in the striatum of Parkinson’s mice were fully restored. That led to a partial recovery of the striatal DAergic and locomotor deficits seen in the chronic Parkinson’s mice. These results imply that long-term melatonin is mitochondrial and moderately neuronal protective in chronic Parkinson’s mice. Melatonin may potentially be effective for slowing down the progression of idiopathic PD and for reducing oxidative stress and respiratory chain inhibition.

**Table 4 ijms-24-03022-t004:** Effects of melatonin on oxidative stress in animal PD models.

PD Experimental Model	Melatonin (Dose and Administration Route	Effects of Melatonin	References
6-OHDA injections into the medial forebrain bundle (Wistar rats)	10 mg/kg, i.p	Reduced oxidative damage and apoptosis of DA neurons	[143]
6-OHDA injections into the medial forebrain bundle (Wistar rats)	10 mg/kg, a day i.p	Improved DA neurons against antioxidant enzyme activities and reduced lipid peroxidation	[142]
MPP+ SNc injection (Sprague-Dawley rats)	10 mg/kg, i.p	Reduced lipid peroxidation and protected against DA neuronal loss induced by MPP+	[145]
MPP+ SNc injections (Wistar rats)	10 mg/kg, i.p	Decreased MPP+-induced toxicity and recovered GSH levels	[146]
MPTP administration in mice	10 mg/kg, i.p	scavenged ·OH, increased GSH and cytosolic SOD in neuronal perikarya	[147]
MPTP injection (C57BL/6 mice)	20 mg/kg, s.c	Reduced mitochondrial NO levels, reduced lipid peroxidation and improved complex I activity in striatum and SNc	[148]
MPTP injections for 18 weeks (C57BL/6 mice)	5 mg/kg, /day i.p	Reduced DA neuronal loss and locomotor activity deficits. Improved mitochondrial respiration, ATP production, and antioxidant enzyme levels in SNc	[144]
MPTP injected in two doses in C57BL/6 mice	20 mg/kg, s.c	Decreasing the outcome of lipid peroxides products and nitrite/nitrate levels significantly	[149]
MPTP injections (swiss mice)	5 or 10 mg/kg/day, p.o	Improved motor performance, striatal DA level, GSH, and antioxidant enzyme activities, and reduced lipid peroxidation. Improved motor response to L-DOPA	[150]
C57BL/6 mice receiving MPTP	10 mg/kg s.c	Preserved mitochondrial oxygen consumption, increased NOS activity and reduced locomotor activity	[151]
Hcy rat model of PD (Sprague-Dawley rats)	10, 20, or 30 mg/kg, i.p	Improved mitochondrial complex-I activity, scavenging of ·OH, restoration of GSH level and elevation activity of SOD and CAT	[148]
Rotenone nigral injection (Sprague-Dawley rats)	10, 20, or 30 mg/kg, i.p	Reduced levels of hydroxyl radicals in mitochondria and increased GSH levels and antioxidant enzymes activities in SNc	[152]
maneb (MB)- and paraquat (PQ)-induced PD model (Male Swiss albino mice)	MB (30 mg/kg) and PQ (10 mg/kg), twice a week for 9 wk	Decreased lipid peroxidation and the number of degenerating neurons	[153]

Jin et al. [145] (1998) reported that melatonin decreased MDA in a rat model of PD induced by MPTP. Melatonin administered at 10 mg/kg increased the GSSG/GSH ratio and increased GSH in the brain in a rat model of PD induced by MPP+ [146]. Thomas and Mohanakumar [149] (2004) reported that melatonin led to scavenged ·OH formed, increased GSH, and an increase in the activity of cytosolic SOD in neuronal perikarya of PD mice. Another study also proved that melatonin significantly decreased oxidative stress caused by MPTP-induced PD in mice [150]. More recent studies using MPTP models for treatment with melatonin have established improved scavenging of ·OH, antioxidant activity, and motor performance [148,151,154]. Similar data indicate activation of the antioxidant system in the Hcy rat model, rotenone, and maneb- and paraquat-induced PD models when treated with melatonin [152,153]. A novel study [155] conducted the combined treatment with melatonin and DAergic neuron transplantation of an animal PD model induced by 6-OHDA, which showed an increased number of neurons in SN and striatum and an increased level of GSH content in treatment groups.
In vitro study


Several cell systems have been developed to study the pathogenesis of PD and treatment with different drugs, including human embryonic kidney 293 (HEK293) cells, human neuroglioma(H4) cells, pheochromocytoma (PC12) cells derived from the rat adrenal medulla [156]. Widely used in cell systems is the human neuroblastoma cell line SH-SY5Y because of its dopaminergic phenotype, which is typical of PD pathology. 

Cellular toxicity induced by 6-OHDA is manifested by increased free-radical production. For this reason, the antioxidant properties of melatonin presumably account for its ability to suppress both necrosis and apoptosis by preventing the reduction of mRNAs for several antioxidant enzymes such as CuZn- SOD, Mn-SOD, and GPx, which normally occur several hours after cells are incubated with 6-OHDA. [157]. Jung et al. [158] (2022) reported that melatonin successfully reduced the neurodegeneration in MPP+ treated retinoic acid differentiated SH-SY5Y cells, improving the antioxidant enzyme activities and reducing the lipid peroxidation via HSF1/HSP70 pathway.

A study suggests melatonin’s role in mitochondrial homeostasis as a powerful hydroxyl radical (·OH) scavenger in vitro in mitochondrial P2 fractions obtained from the cerebral cortex of an MPP+-induced PD animal model [147]. 

A study by Zneng et al. [159] (2021) showed that melatonin significantly decreased ROS production and suppressed NLRP3 inflammasome activity in mouse BV2 cells and primary microglial cells pre-treated with MPP (+) (Table 5). Similarly, Panyada et al. [160] (2015) showed that melatonin significantly protected neuron cells from the toxicity of ROS occurring from serum deprivation-induced oxidative stress.

Both direct and indirect antioxidant actions are thought to be important mediators of melatonin’s antiparkinsonian.

### 2.3. Clinical Data Supporting the Role of Enormous Oxidative Stress in the Pathogenesis of PD

Knowledge gained about the link between oxidative stress and the pathogenesis of PD has two therapeutic implications. The first is whether currently used drugs have any impact on this process, either positive or negative. The second is the potential to develop new drugs that might be able to mitigate oxidative stress and, by doing so, slow disease progression. Extensive investigations into the first question have shown no solid evidence for clinically relevant impact, and the success of the second scenario remains elusive. PD remains an incurable disease despite repeated efforts to test potential neuroprotective strategies. Although most of the clinical trials so far performed with antioxidants did not reach the expected results, with some exceptions, some new drugs have been described as possessing very interesting properties that might be exploited in the context of PD therapy. Among them, melatonin is probably the most promising molecule because of its amphipathic nature and ability to easily cross the BBB and inhibit the occurrence of oxidative stress. However, due to the lack of long-term clinical observations, its therapeutic effect remains skeptical.

In spite of many studies carried out on animal models, only a few clinical trials have been performed to assess the therapeutic potential of melatonin in PD patients, and they were mostly focused on sleep disorders associated with PD pathology. There are only several clinical studies that evaluated the antioxidant effect of melatonin on patients with PD. 

A double-blind clinical trial was accomplished in 26 patients, divided into 2 groups and treated with melatonin (25 mg) and placebo, respectively, at noon and 30 min before bedtime for 90 days. The group treated with melatonin demonstrated a significantly reduced plasma level of lipoperoxides, nitric oxide metabolites, and carbonyl groups in comparison with the placebo group. Conversely, CAT activity was increased significantly in comparison with the placebo group. Compared with the placebo group, the melatonin group showed significant increases in mitochondrial complex activity and respiratory control ratio. [161]. The supplementation with melatonin had a powerful antioxidant effect via correction of the rate of complex I activity and respiratory control ratio. These results suggest that melatonin could play a role in the treatment of PD.

A double-blind, placebo-controlled clinical trial was conducted among 60 patients with PD. Participants were randomly divided into 2 groups to intake either 10 mg melatonin (two melatonin capsules, 5 mg each) (n = 30) or a placebo (n = 30) once a day, 1 h before bedtime, for 12 weeks. Melatonin supplementation significantly increased plasma total antioxidant capacity and total GSH levels [162]. Two considerations can emerge from the clinical trials mentioned above. First, melatonin is safe and well-tolerated in humans. Second, the small size of the clinical trials conducted until now does not allow us to achieve any conclusion on the therapeutic potential of melatonin, pointing out the need to design and perform a large-scale study.

## 3. Conclusions

The elucidation of the complex interrelation between oxidative stress and symptoms of AD and PD is essential for the design of adequate and relevant timed therapeutic approaches. Disturbed homeostasis of pro-oxidant/antioxidant endogenous components can be early markers, making the brain vulnerable to initiation and progressive development of these neurodegenerative diseases. Notably, aging facilitates oxidative damage and the most common form of AD, dementia, from one side, and impaired motor coordination and muscle tone in PD. In this context, preclinical and clinical studies are focused on elucidating the molecular mechanisms underlying the role of excessive oxidative stress in the etiology of these diseases (Figure 1). This effort should contribute to creating more precise prophylactic and attenuation strategies for the further progression of these irreversible diseases. A common feature of the pathogenesis of AD and PD is the impaired circadian rhythms associated with a decreased release of the hormone melatonin from the pineal gland [70,163,164]. Although there is an evaluation of the impact of oxidative stress on the progression of the two diseases, as well as numerous experimental studies exploring the beneficial effect of melatonin treatment as a putative antioxidant, a crucial issue in experimental design is the period when should begin the treatment with melatonin. Literature data suggest that, although melatonin supplementation is ineffective in old transgenic mice, this antioxidant can prevent extracellular Aβ deposition when the treatment is applied as a prophylactic procedure [73,74,75,140,141,142]. Therefore, more reliable results are expected with pretreatment drug protocols in middle-aged transgenic models or old intact animals before the inducement of the pathogenesis in AD and PD models.

## Figures and Tables

**Figure 1 ijms-24-03022-f001:**
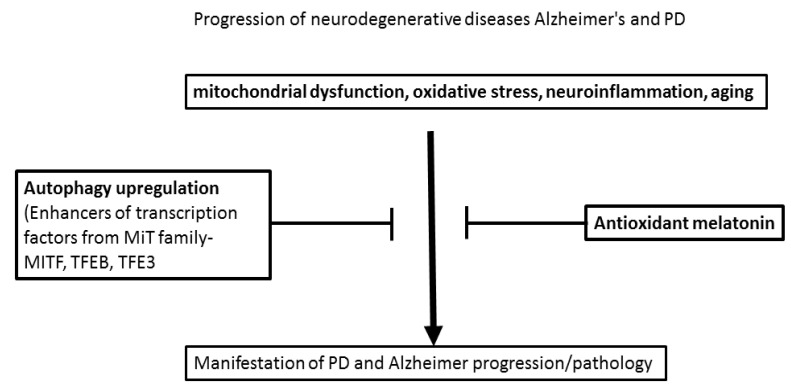
Molecular mechanisms which can reverse or delay the development of Alzheimer and PD pathology.

**Table 1 ijms-24-03022-t001:** In vitro and in vivo models considering the link between oxidative stress, beta-amyloid (Aβ) peptide, and hyperphosphorylation of tau protein (p-Tau).

In Vitro and In Vivo Models	Oxidative Stress	Beta-Amyloid	p-Tau	References
cerebral endothelial cell culture from transgenic Tg2576 mouse	1 μM H_2_O_2_ exposure for up to 48 h	VEGF↑, sAPPβ↑ and sAPPα ↑ present in the medium	-	[19]
Deletion of PGE2 EP2 receptor in the APPSwe-PS1E9 mice (female aged eight months, male of 12 months	F2-isoprostane, a marker of docosohexanoic acid oxidation in the brain↑	BACE1↑ Aβ_1–42_ ↑ Aβ_1–40_ ↑	-	[21]
Male Long-Evans rats (3–4 months) infused for four weeks with Aβ + pro-oxidants	Fe^2+^, or buthionine-sulfoximine treatment simultaneously with Aβ	Aβ exposure	p-Tau in CSF ↑	[22]
Female triple-transgenic mouse model of AD (3–5 months)	GSH and vitamin E ↓; lipid peroxidation ↑; GPx and SOD ↑	Aβ ↑ after oxidative stress	p-Tau ↑ after oxidative stress	[23]
PC12 cells	superoxide dismutase (SOD) and hemoxygenase-i (HO^−1^) ↑	Aβ_25–35_ exposure	-	[24]
Male transgenic mouse model of AD (4-month-old and 21 to 25 months)	CuZn SOD and HO^−1^ ↑ in aged mice	Aβ deposits in aged mice	-	[24]
Wistar male rats treated with aluminum from 12 months to 26-months	HNE, a marker for oxidative damage in the hippocampus ↑	-	p-Tau in the hippocampus ↑	[25]
Male transgenic mice (Tg2576)	Isoprostane, a marker of in vivo oxidative stress	Aβ↑	-	[26]
Cell cultures (hippocampus and glia) from Sprague Dawley rat pups 2–4 d postpartum	Oxidative stress-induced changes in mitochondrial potential in glia	Aβ exposure	-	[27]
pin + icvAβ_1–42_ rat model of AD	GSH ↓; SOD and MDA ↑ in the hippocampus	Aβ↑	-	[28]
double transgenic B6.Cg-Tg (APPswe, PSENdE9)85Dbo/Mmjax (APP/PS1) AD-model mice aged ten months	MDA ↑, GSH ↓	-	-	[29]
Neurons in the brain of a mouse model of AD that contains mutant human amyloid precursor protein and presenilin 1	Oxidative stress (detected via changed redox potential)	Aβ ↑ near regions with increased oxidative stress	-	[30]
Male transgenic mice (C57B61 SJL x FVB or C57B6/SJL; 13—25 months old)	Markers of oxidative lipid and protein damage↑	Aβ↑	-	[31]

**Table 2 ijms-24-03022-t002:** Clinical data considering the link between oxidative stress, beta-amyloid peptide, and p-Tau protein.

Human Sample	Melatonin	Oxidative Stress	Aβ Protein	p-Tau	References
Patients with AD (early stage)	-	RNA oxidation in the cortex, hippocampus, amygdala ↓	-	-	[38]
Subjects with CMI	-	Markers of oxidized proteins and lipids in the superior and middle temporal gyri	-	-	[39]
Human neuroblastoma cells	-	H_2_O_2_ medium	H2O2-evoked Aβ ↑	-	[40]
Down syndrome	-	oxidized nucleic acid ↑, 8-hydroxyguanosine (8OHG) ↑, and oxidized protein, nitrotyrosine ↑ in cerebral neurons		-	[41]
Patients with AD (early stage)	-	Marker of oxidized RNA in CSF ↑; in serum ↓	-	-	[42]
Subjects with CMI and patients with AD (early stage)	-	Oxidative DNA damage in leukocytes	-	-	[43]
Human brain tissue and pineal gland	CSF melatonin levels	-	Aβ ↑	-	[51]

**Table 5 ijms-24-03022-t005:** Effects of melatonin on oxidative stress in in vitro PD models.

In Vitro PD Experimental Model	Melatonin Concentration	Effects of Melatonin	References
MPP+-treated mitochondrial fraction	0.1–100 μM	Reduction of •OH formation	[147]
6-OHDA-treated PC12 cells	1–100 nM	Protection from both apoptosis and necrosis, increasing antioxidant enzymes	[156]
MPP+ treated retinoic acid-differentiated SH-SY5Y cells	100 μM	Improved antioxidant enzyme activities and reduced lipid peroxidation	[158]
MPP+ treated mouse BV2 cells and primary microglial cells	100 μM	Reduced ROS production and suppress NLRP3 inflammasome activity	[159]
P19-derived neurons	10 μM	Rescuing of dopamine neurons from spontaneous cell death in low-density seeding cultures	[160]

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
