# Peer review of "Oxidative Stress and Aging as Risk Factors for Alzheimer’s Disease and Parkinson’s Disease: The Role of the Antioxidant Melatonin"

_ijms, 2023, doi:10.3390/ijms24033022_

Round 1

Reviewer 1 Report

The authors in this review article determined “Oxidative stress and aging as risk factors for Alzheimer’s disease and Parkinson's disease: the role of the antioxidant melatonin”. The current study demonstrated the Aging and neurodegenerative diseases share common hallmarks, including mitochondrial dysfunction and protein aggregation. Moreover, one of the major issues of the demographic crisis today is related to the progressive rise in costs for care and maintenance of the standard living condition of aged patients with neurodegenerative diseases. There is a divergence in the etiology of neurodegenerative diseases. Still, a disturbed endogenous pro-oxidants/antioxidants balance is considered the crucial detrimental factor that makes the brain vulnerable to aging and progressive neurodegeneration. The present review focuses on the complex relationships between oxidative stress and the two most frequent neurodegenerative diseases associated with aging, Alzheimer's disease (ADs) and Parkinson's disease (PD). Most of the available data support the hypothesis that a disturbed antioxidant defense system is a prerequisite for developing pathogenesis and clinical symptoms of ADs and PD. Furthermore, the release of the endogenous hormone melatonin from the pineal gland progressively diminishes with aging, while the susceptibility of people to these diseases increases with age. Elucidation of the underlying mechanisms involved in deleterious conditions predisposing to neurodegeneration in aging, including the diminished role of melatonin, is important for elaborating precise treatment strategies for the pathogenesis in AD and PD.

I can see very few articles in the present topic, which adds advantage for this study to be novel even though there are some flaws in the literature collection and sub sections of the article. I would like to recommend some minor concerns to the authors to fulfil the hypothesis, as there are scarce citations of recent studies in the article.

1.    The authors claim regulation of melatonin can be used as an effective strategy in AD and PD therapy, But the authors should not ignore the main targets of Oxidative stress and autophagy, namely TFEB and TFE3 which are the master regulators of autophagy and lysosomal biogenesis, which would add some novelty.

2.    Introduction about the various symptoms of AD and PD need to be in detail, like tau pathology, Autophagy impairment and other degradative pathways involved in Oxidative stress and tau pathology. https://doi.org/10.1186/s12929-022-00871-6, https://doi.org/10.2174/1874467214666210906125318.

3.    In the text and table, ‘ptau and pTau” need to be changed as one type either p-Tau or pTau, kindly correct.

4.    In subtopic ‘The role of enormous oxidative stress in triggering Aβ accumulation and p-tau’ (in vitro and in vivo models) is not clearly explained, can add few transgenic mice models which shows ROS production and oxidative stress in AD in vivo models, please refer https://doi.org/10.3389/fmolb.2022.1050768, https://doi.org/10.3389/fmolb.2022.1030534.

5.    TFEB is one of the main modulators of Oxidative stress and improve melatonin functions refer, https://doi.org/10.1038/s41401-021-00711-7.

6.    The authors need to check the abbreviations, in first time usage they need to use the full word and abbreviation. A separate section for abbreviation would be better.

7.    The authors need to show the final schematic conclusion diagram revealing a summarized concluding evidence and mechanism.

8.    A careful English check and grammatical errors need to be resolved by the authors.

The available research information seems to be insufficient, and the authors need to address the above comments. Taking together to all this issue I recommend minor revision to the manuscript in present form.

Author Response

We are thankful for your critical review and your valuable suggestions. Indeed corrections and modifications are helpful for the optimization of the manuscript. We hereby addressed your raised comments point-by-point basis on the followi.

Reviewer' 1

Point # 1.    The authors claim regulation of melatonin can be used as an effective strategy in AD and PD therapy, But the authors should not ignore the main targets of Oxidative stress and autophagy, namely TFEB and TFE3 which are the master regulators of autophagy and lysosomal biogenesis, which would add some novelty.

Response: We are thankful for this suggestion. We added data from recent studies for the protective role of melatonin that is closely related to TFEB and TFE3 factors in the pathogenesis of AD and PD (see in subsection 1.1. and 2.).

Point # 2.    Introduction about the various symptoms of AD and PD need to be in detail, like tau pathology, Autophagy impairment and other degradative pathways involved in Oxidative stress and tau pathology. https://doi.org/10.1186/s12929-022-00871-6, https://doi.org/10.2174/1874467214666210906125318.

Response: We’re thankful for this relevant note of the Reviewer. We decided to arrange the text with more details of degradative pathways closely associated with OS and the two diseases in the subsections with the effects of melatonin. Thus, when we mentioned some effect of melatonin treatment on AD or PD and some mechanism related to the pathogenesis was affected we discussed this pathway and how it was involved in either AD or PD.

Point # 3.   In the text and table, ‘ptau and pTau” need to be changed as one type either p-Tau or pTau, kindly correct.

Response: Corrected. Thank you for the note!

Point # 4.   In subtopic ‘The role of enormous oxidative stress in triggering Aβ accumulation and p-tau’ (in vitro and in vivo models) is not clearly explained, can add few transgenic mice models which shows  ROS production and oxidative stress in AD in vivo models, please refer https://doi.org/10.3389/fmolb.2022.1050768, https://doi.org/10.3389/fmolb.2022.1030534.

Response: In the subsection entitled:The role of enormous oxidative stress in triggering Aβ accumulation and p-Tau (in vitro and in vivo models)” we involved only data closely related to the topic with an idea to support with available findings on this topic that abnormally elevated oxidative stress might triggers  Aβ deposition and induces pathogenesis of AD.

Point # 5.   TFEB is one of the main modulators of Oxidative stress and improve melatonin functions refer, https://doi .org/10.1038/s41401-021-00711-7.

Response: The key role of TFEB is mentioned in the new version as well as the effect of melatonin on this important to mitophagy signaling molecule in AD.

Point # 6.   The authors need to check the abbreviations, in first time usage they need to use the full word and abbreviation. A separate  section for abbreviation would be better.

Response: We’re grateful for this recommendation of the reviewer and checked carefully when to start usage of abbreviation (after the first mention). A separate section of abbreviation is also included after the Abstract.

Point #7.    The authors need to show the final schematic conclusion diagram revealing a summarized concluding evidence and  mechanism.

Response: The schematic diagram in the conclusion was also prepared and inserted in the new version in Conclusion section.

Point #8.    A careful English check and grammatical errors need to be resolved  by the authors.

Response: We used paid Grammar software again to check and correct if it was necessary all grammatical errors. 

We are thankful for your critical review and your valuable suggestions. Indeed corrections and modifications are helpful for the optimization of the manuscript. We hereby addressed your raised comments point-by-point basis on the followi.

Reviewer' 1

Point # 1.    The authors claim regulation of melatonin can be used as an effective strategy in AD and PD therapy, But the authors should not ignore the main targets of Oxidative stress and autophagy, namely TFEB and TFE3 which are the master regulators of autophagy and lysosomal biogenesis, which would add some novelty.

Response: We are thankful for this suggestion. We added data from recent studies for the protective role of melatonin that is closely related to TFEB and TFE3 factors in the pathogenesis of AD and PD (see in subsection 1.1. and 2.).

Point # 2.    Introduction about the various symptoms of AD and PD need to be in detail, like tau pathology, Autophagy impairment and other degradative pathways involved in Oxidative stress and tau pathology. https://doi.org/10.1186/s12929-022-00871-6, https://doi.org/10.2174/1874467214666210906125318.

Response: We’re thankful for this relevant note of the Reviewer. We decided to arrange the text with more details of degradative pathways closely associated with OS and the two diseases in the subsections with the effects of melatonin. Thus, when we mentioned some effect of melatonin treatment on AD or PD and some mechanism related to the pathogenesis was affected we discussed this pathway and how it was involved in either AD or PD.

Point # 3.   In the text and table, ‘ptau and pTau” need to be changed as one type either p-Tau or pTau, kindly correct.

Response: Corrected. Thank you for the note!

Point # 4.   In subtopic ‘The role of enormous oxidative stress in triggering Aβ accumulation and p-tau’ (in vitro and in vivo models) is not clearly explained, can add few transgenic mice models which shows  ROS production and oxidative stress in AD in vivo models, please refer https://doi.org/10.3389/fmolb.2022.1050768, https://doi.org/10.3389/fmolb.2022.1030534.

Response: In the subsection entitled:The role of enormous oxidative stress in triggering Aβ accumulation and p-Tau (in vitro and in vivo models)” we involved only data closely related to the topic with an idea to support with available findings on this topic that abnormally elevated oxidative stress might triggers  Aβ deposition and induces pathogenesis of AD.

Point # 5.   TFEB is one of the main modulators of Oxidative stress and improve melatonin functions refer, https://doi .org/10.1038/s41401-021-00711-7.

Response: The key role of TFEB is mentioned in the new version as well as the effect of melatonin on this important to mitophagy signaling molecule in AD.

Point # 6.   The authors need to check the abbreviations, in first time usage they need to use the full word and abbreviation. A separate  section for abbreviation would be better.

Response: We’re grateful for this recommendation of the reviewer and checked carefully when to start usage of abbreviation (after the first mention). A separate section of abbreviation is also included after the Abstract.

Point #7.    The authors need to show the final schematic conclusion diagram revealing a summarized concluding evidence and  mechanism.

Response: The schematic diagram in the conclusion was also prepared and inserted in the new version in Conclusion section.

Point #8.    A careful English check and grammatical errors need to be resolved  by the authors.

Response: We used paid Grammar software again to check and correct if it was necessary all grammatical errors. 

We are thankful for your critical review and your valuable suggestions. Indeed corrections and modifications are helpful for the optimization of the manuscript. We hereby addressed your raised comments point-by-point basis on the followi.

Reviewer' 1

Point # 1.    The authors claim regulation of melatonin can be used as an effective strategy in AD and PD therapy, But the authors should not ignore the main targets of Oxidative stress and autophagy, namely TFEB and TFE3 which are the master regulators of autophagy and lysosomal biogenesis, which would add some novelty.

Response: We are thankful for this suggestion. We added data from recent studies for the protective role of melatonin that is closely related to TFEB and TFE3 factors in the pathogenesis of AD and PD (see in subsection 1.1. and 2.).

Point # 2.    Introduction about the various symptoms of AD and PD need to be in detail, like tau pathology, Autophagy impairment and other degradative pathways involved in Oxidative stress and tau pathology. https://doi.org/10.1186/s12929-022-00871-6, https://doi.org/10.2174/1874467214666210906125318.

Response: We’re thankful for this relevant note of the Reviewer. We decided to arrange the text with more details of degradative pathways closely associated with OS and the two diseases in the subsections with the effects of melatonin. Thus, when we mentioned some effect of melatonin treatment on AD or PD and some mechanism related to the pathogenesis was affected we discussed this pathway and how it was involved in either AD or PD.

Point # 3.   In the text and table, ‘ptau and pTau” need to be changed as one type either p-Tau or pTau, kindly correct.

Response: Corrected. Thank you for the note!

Point # 4.   In subtopic ‘The role of enormous oxidative stress in triggering Aβ accumulation and p-tau’ (in vitro and in vivo models) is not clearly explained, can add few transgenic mice models which shows  ROS production and oxidative stress in AD in vivo models, please refer https://doi.org/10.3389/fmolb.2022.1050768, https://doi.org/10.3389/fmolb.2022.1030534.

Response: In the subsection entitled:The role of enormous oxidative stress in triggering Aβ accumulation and p-Tau (in vitro and in vivo models)” we involved only data closely related to the topic with an idea to support with available findings on this topic that abnormally elevated oxidative stress might triggers  Aβ deposition and induces pathogenesis of AD.

Point # 5.   TFEB is one of the main modulators of Oxidative stress and improve melatonin functions refer, https://doi .org/10.1038/s41401-021-00711-7.

Response: The key role of TFEB is mentioned in the new version as well as the effect of melatonin on this important to mitophagy signaling molecule in AD.

Point # 6.   The authors need to check the abbreviations, in first time usage they need to use the full word and abbreviation. A separate  section for abbreviation would be better.

Response: We’re grateful for this recommendation of the reviewer and checked carefully when to start usage of abbreviation (after the first mention). A separate section of abbreviation is also included after the Abstract.

Point #7.    The authors need to show the final schematic conclusion diagram revealing a summarized concluding evidence and  mechanism.

Response: The schematic diagram in the conclusion was also prepared and inserted in the new version in Conclusion section.

Point #8.    A careful English check and grammatical errors need to be resolved  by the authors.

Response: We used paid Grammar software again to check and correct if it was necessary all grammatical errors. 

We are thankful for your critical review and your valuable suggestions. Indeed corrections and modifications are helpful for the optimization of the manuscript. We hereby addressed your raised comments point-by-point basis on the followi.

Reviewer' 1

Point # 1.    The authors claim regulation of melatonin can be used as an effective strategy in AD and PD therapy, But the authors should not ignore the main targets of Oxidative stress and autophagy, namely TFEB and TFE3 which are the master regulators of autophagy and lysosomal biogenesis, which would add some novelty.

Response: We are thankful for this suggestion. We added data from recent studies for the protective role of melatonin that is closely related to TFEB and TFE3 factors in the pathogenesis of AD and PD (see in subsection 1.1. and 2.).

Point # 2.    Introduction about the various symptoms of AD and PD need to be in detail, like tau pathology, Autophagy impairment and other degradative pathways involved in Oxidative stress and tau pathology. https://doi.org/10.1186/s12929-022-00871-6, https://doi.org/10.2174/1874467214666210906125318.

Response: We’re thankful for this relevant note of the Reviewer. We decided to arrange the text with more details of degradative pathways closely associated with OS and the two diseases in the subsections with the effects of melatonin. Thus, when we mentioned some effect of melatonin treatment on AD or PD and some mechanism related to the pathogenesis was affected we discussed this pathway and how it was involved in either AD or PD.

Point # 3.   In the text and table, ‘ptau and pTau” need to be changed as one type either p-Tau or pTau, kindly correct.

Response: Corrected. Thank you for the note!

Point # 4.   In subtopic ‘The role of enormous oxidative stress in triggering Aβ accumulation and p-tau’ (in vitro and in vivo models) is not clearly explained, can add few transgenic mice models which shows  ROS production and oxidative stress in AD in vivo models, please refer https://doi.org/10.3389/fmolb.2022.1050768, https://doi.org/10.3389/fmolb.2022.1030534.

Response: In the subsection entitled:The role of enormous oxidative stress in triggering Aβ accumulation and p-Tau (in vitro and in vivo models)” we involved only data closely related to the topic with an idea to support with available findings on this topic that abnormally elevated oxidative stress might triggers  Aβ deposition and induces pathogenesis of AD.

Point # 5.   TFEB is one of the main modulators of Oxidative stress and improve melatonin functions refer, https://doi .org/10.1038/s41401-021-00711-7.

Response: The key role of TFEB is mentioned in the new version as well as the effect of melatonin on this important to mitophagy signaling molecule in AD.

Point # 6.   The authors need to check the abbreviations, in first time usage they need to use the full word and abbreviation. A separate  section for abbreviation would be better.

Response: We’re grateful for this recommendation of the reviewer and checked carefully when to start usage of abbreviation (after the first mention). A separate section of abbreviation is also included after the Abstract.

Point #7.    The authors need to show the final schematic conclusion diagram revealing a summarized concluding evidence and  mechanism.

Response: The schematic diagram in the conclusion was also prepared and inserted in the new version in Conclusion section.

Point #8.    A careful English check and grammatical errors need to be resolved  by the authors.

Response: We used paid Grammar software again to check and correct if it was necessary all grammatical errors. 

We are thankful for your critical review and your valuable suggestions. Indeed corrections and modifications are helpful for the optimization of the manuscript. We hereby addressed your raised comments point-by-point basis on the followi.

Reviewer' 1

Point # 1.    The authors claim regulation of melatonin can be used as an effective strategy in AD and PD therapy, But the authors should not ignore the main targets of Oxidative stress and autophagy, namely TFEB and TFE3 which are the master regulators of autophagy and lysosomal biogenesis, which would add some novelty.

Response: We are thankful for this suggestion. We added data from recent studies for the protective role of melatonin that is closely related to TFEB and TFE3 factors in the pathogenesis of AD and PD (see in subsection 1.1. and 2.).

Point # 2.    Introduction about the various symptoms of AD and PD need to be in detail, like tau pathology, Autophagy impairment and other degradative pathways involved in Oxidative stress and tau pathology. https://doi.org/10.1186/s12929-022-00871-6, https://doi.org/10.2174/1874467214666210906125318.

Response: We’re thankful for this relevant note of the Reviewer. We decided to arrange the text with more details of degradative pathways closely associated with OS and the two diseases in the subsections with the effects of melatonin. Thus, when we mentioned some effect of melatonin treatment on AD or PD and some mechanism related to the pathogenesis was affected we discussed this pathway and how it was involved in either AD or PD.

Point # 3.   In the text and table, ‘ptau and pTau” need to be changed as one type either p-Tau or pTau, kindly correct.

Response: Corrected. Thank you for the note!

Point # 4.   In subtopic ‘The role of enormous oxidative stress in triggering Aβ accumulation and p-tau’ (in vitro and in vivo models) is not clearly explained, can add few transgenic mice models which shows  ROS production and oxidative stress in AD in vivo models, please refer https://doi.org/10.3389/fmolb.2022.1050768, https://doi.org/10.3389/fmolb.2022.1030534.

Response: In the subsection entitled:The role of enormous oxidative stress in triggering Aβ accumulation and p-Tau (in vitro and in vivo models)” we involved only data closely related to the topic with an idea to support with available findings on this topic that abnormally elevated oxidative stress might triggers  Aβ deposition and induces pathogenesis of AD.

Point # 5.   TFEB is one of the main modulators of Oxidative stress and improve melatonin functions refer, https://doi .org/10.1038/s41401-021-00711-7.

Response: The key role of TFEB is mentioned in the new version as well as the effect of melatonin on this important to mitophagy signaling molecule in AD.

Point # 6.   The authors need to check the abbreviations, in first time usage they need to use the full word and abbreviation. A separate  section for abbreviation would be better.

Response: We’re grateful for this recommendation of the reviewer and checked carefully when to start usage of abbreviation (after the first mention). A separate section of abbreviation is also included after the Abstract.

Point #7.    The authors need to show the final schematic conclusion diagram revealing a summarized concluding evidence and  mechanism.

Response: The schematic diagram in the conclusion was also prepared and inserted in the new version in Conclusion section.

Point #8.    A careful English check and grammatical errors need to be resolved  by the authors.

Response: We used paid Grammar software again to check and correct if it was necessary all grammatical errors. 

We are thankful for your critical review and your valuable suggestions. Indeed corrections and modifications are helpful for the optimization of the manuscript. We hereby addressed your raised comments point-by-point basis on the followi.

Reviewer' 1

Point # 1.    The authors claim regulation of melatonin can be used as an effective strategy in AD and PD therapy, But the authors should not ignore the main targets of Oxidative stress and autophagy, namely TFEB and TFE3 which are the master regulators of autophagy and lysosomal biogenesis, which would add some novelty.

Response: We are thankful for this suggestion. We added data from recent studies for the protective role of melatonin that is closely related to TFEB and TFE3 factors in the pathogenesis of AD and PD (see in subsection 1.1. and 2.).

Point # 2.    Introduction about the various symptoms of AD and PD need to be in detail, like tau pathology, Autophagy impairment and other degradative pathways involved in Oxidative stress and tau pathology. https://doi.org/10.1186/s12929-022-00871-6, https://doi.org/10.2174/1874467214666210906125318.

Response: We’re thankful for this relevant note of the Reviewer. We decided to arrange the text with more details of degradative pathways closely associated with OS and the two diseases in the subsections with the effects of melatonin. Thus, when we mentioned some effect of melatonin treatment on AD or PD and some mechanism related to the pathogenesis was affected we discussed this pathway and how it was involved in either AD or PD.

Point # 3.   In the text and table, ‘ptau and pTau” need to be changed as one type either p-Tau or pTau, kindly correct.

Response: Corrected. Thank you for the note!

Point # 4.   In subtopic ‘The role of enormous oxidative stress in triggering Aβ accumulation and p-tau’ (in vitro and in vivo models) is not clearly explained, can add few transgenic mice models which shows  ROS production and oxidative stress in AD in vivo models, please refer https://doi.org/10.3389/fmolb.2022.1050768, https://doi.org/10.3389/fmolb.2022.1030534.

Response: In the subsection entitled:The role of enormous oxidative stress in triggering Aβ accumulation and p-Tau (in vitro and in vivo models)” we involved only data closely related to the topic with an idea to support with available findings on this topic that abnormally elevated oxidative stress might triggers  Aβ deposition and induces pathogenesis of AD.

Point # 5.   TFEB is one of the main modulators of Oxidative stress and improve melatonin functions refer, https://doi .org/10.1038/s41401-021-00711-7.

Response: The key role of TFEB is mentioned in the new version as well as the effect of melatonin on this important to mitophagy signaling molecule in AD.

Point # 6.   The authors need to check the abbreviations, in first time usage they need to use the full word and abbreviation. A separate  section for abbreviation would be better.

Response: We’re grateful for this recommendation of the reviewer and checked carefully when to start usage of abbreviation (after the first mention). A separate section of abbreviation is also included after the Abstract.

Point #7.    The authors need to show the final schematic conclusion diagram revealing a summarized concluding evidence and  mechanism.

Response: The schematic diagram in the conclusion was also prepared and inserted in the new version in Conclusion section.

Point #8.    A careful English check and grammatical errors need to be resolved  by the authors.

Response: We used paid Grammar software again to check and correct if it was necessary all grammatical errors. 

We are thankful for your critical review and your valuable suggestions. Indeed corrections and modifications are helpful for the optimization of the manuscript. We hereby addressed your raised comments point-by-point basis on the followi.

Reviewer' 1

Point # 1.    The authors claim regulation of melatonin can be used as an effective strategy in AD and PD therapy, But the authors should not ignore the main targets of Oxidative stress and autophagy, namely TFEB and TFE3 which are the master regulators of autophagy and lysosomal biogenesis, which would add some novelty.

Response: We are thankful for this suggestion. We added data from recent studies for the protective role of melatonin that is closely related to TFEB and TFE3 factors in the pathogenesis of AD and PD (see in subsection 1.1. and 2.).

Point # 2.    Introduction about the various symptoms of AD and PD need to be in detail, like tau pathology, Autophagy impairment and other degradative pathways involved in Oxidative stress and tau pathology. https://doi.org/10.1186/s12929-022-00871-6, https://doi.org/10.2174/1874467214666210906125318.

Response: We’re thankful for this relevant note of the Reviewer. We decided to arrange the text with more details of degradative pathways closely associated with OS and the two diseases in the subsections with the effects of melatonin. Thus, when we mentioned some effect of melatonin treatment on AD or PD and some mechanism related to the pathogenesis was affected we discussed this pathway and how it was involved in either AD or PD.

Point # 3.   In the text and table, ‘ptau and pTau” need to be changed as one type either p-Tau or pTau, kindly correct.

Response: Corrected. Thank you for the note!

Point # 4.   In subtopic ‘The role of enormous oxidative stress in triggering Aβ accumulation and p-tau’ (in vitro and in vivo models) is not clearly explained, can add few transgenic mice models which shows  ROS production and oxidative stress in AD in vivo models, please refer https://doi.org/10.3389/fmolb.2022.1050768, https://doi.org/10.3389/fmolb.2022.1030534.

Response: In the subsection entitled:The role of enormous oxidative stress in triggering Aβ accumulation and p-Tau (in vitro and in vivo models)” we involved only data closely related to the topic with an idea to support with available findings on this topic that abnormally elevated oxidative stress might triggers  Aβ deposition and induces pathogenesis of AD.

Point # 5.   TFEB is one of the main modulators of Oxidative stress and improve melatonin functions refer, https://doi .org/10.1038/s41401-021-00711-7.

Response: The key role of TFEB is mentioned in the new version as well as the effect of melatonin on this important to mitophagy signaling molecule in AD.

Point # 6.   The authors need to check the abbreviations, in first time usage they need to use the full word and abbreviation. A separate  section for abbreviation would be better.

Response: We’re grateful for this recommendation of the reviewer and checked carefully when to start usage of abbreviation (after the first mention). A separate section of abbreviation is also included after the Abstract.

Point #7.    The authors need to show the final schematic conclusion diagram revealing a summarized concluding evidence and  mechanism.

Response: The schematic diagram in the conclusion was also prepared and inserted in the new version in Conclusion section.

Point #8.    A careful English check and grammatical errors need to be resolved  by the authors.

Response: We used paid Grammar software again to check and correct if it was necessary all grammatical errors. 

We are thankful for your critical review and your valuable suggestions. Indeed corrections and modifications are helpful for the optimization of the manuscript. We hereby addressed your raised comments point-by-point basis on the followi.

Reviewer' 1

Point # 1.    The authors claim regulation of melatonin can be used as an effective strategy in AD and PD therapy, But the authors should not ignore the main targets of Oxidative stress and autophagy, namely TFEB and TFE3 which are the master regulators of autophagy and lysosomal biogenesis, which would add some novelty.

Response: We are thankful for this suggestion. We added data from recent studies for the protective role of melatonin that is closely related to TFEB and TFE3 factors in the pathogenesis of AD and PD (see in subsection 1.1. and 2.).

Point # 2.    Introduction about the various symptoms of AD and PD need to be in detail, like tau pathology, Autophagy impairment and other degradative pathways involved in Oxidative stress and tau pathology. https://doi.org/10.1186/s12929-022-00871-6, https://doi.org/10.2174/1874467214666210906125318.

Response: We’re thankful for this relevant note of the Reviewer. We decided to arrange the text with more details of degradative pathways closely associated with OS and the two diseases in the subsections with the effects of melatonin. Thus, when we mentioned some effect of melatonin treatment on AD or PD and some mechanism related to the pathogenesis was affected we discussed this pathway and how it was involved in either AD or PD.

Point # 3.   In the text and table, ‘ptau and pTau” need to be changed as one type either p-Tau or pTau, kindly correct.

Response: Corrected. Thank you for the note!

Point # 4.   In subtopic ‘The role of enormous oxidative stress in triggering Aβ accumulation and p-tau’ (in vitro and in vivo models) is not clearly explained, can add few transgenic mice models which shows  ROS production and oxidative stress in AD in vivo models, please refer https://doi.org/10.3389/fmolb.2022.1050768, https://doi.org/10.3389/fmolb.2022.1030534.

Response: In the subsection entitled:The role of enormous oxidative stress in triggering Aβ accumulation and p-Tau (in vitro and in vivo models)” we involved only data closely related to the topic with an idea to support with available findings on this topic that abnormally elevated oxidative stress might triggers  Aβ deposition and induces pathogenesis of AD.

Point # 5.   TFEB is one of the main modulators of Oxidative stress and improve melatonin functions refer, https://doi .org/10.1038/s41401-021-00711-7.

Response: The key role of TFEB is mentioned in the new version as well as the effect of melatonin on this important to mitophagy signaling molecule in AD.

Point # 6.   The authors need to check the abbreviations, in first time usage they need to use the full word and abbreviation. A separate  section for abbreviation would be better.

Response: We’re grateful for this recommendation of the reviewer and checked carefully when to start usage of abbreviation (after the first mention). A separate section of abbreviation is also included after the Abstract.

Point #7.    The authors need to show the final schematic conclusion diagram revealing a summarized concluding evidence and  mechanism.

Response: The schematic diagram in the conclusion was also prepared and inserted in the new version in Conclusion section.

Point #8.    A careful English check and grammatical errors need to be resolved  by the authors.

Response: We used paid Grammar software again to check and correct if it was necessary all grammatical errors. 

We are thankful for your critical review and your valuable suggestions. Indeed corrections and modifications are helpful for the optimization of the manuscript. We hereby addressed your raised comments point-by-point basis on the followi.

Reviewer' 1

Point # 1.    The authors claim regulation of melatonin can be used as an effective strategy in AD and PD therapy, But the authors should not ignore the main targets of Oxidative stress and autophagy, namely TFEB and TFE3 which are the master regulators of autophagy and lysosomal biogenesis, which would add some novelty.

Response: We are thankful for this suggestion. We added data from recent studies for the protective role of melatonin that is closely related to TFEB and TFE3 factors in the pathogenesis of AD and PD (see in subsection 1.1. and 2.).

Point # 2.    Introduction about the various symptoms of AD and PD need to be in detail, like tau pathology, Autophagy impairment and other degradative pathways involved in Oxidative stress and tau pathology. https://doi.org/10.1186/s12929-022-00871-6, https://doi.org/10.2174/1874467214666210906125318.

Response: We’re thankful for this relevant note of the Reviewer. We decided to arrange the text with more details of degradative pathways closely associated with OS and the two diseases in the subsections with the effects of melatonin. Thus, when we mentioned some effect of melatonin treatment on AD or PD and some mechanism related to the pathogenesis was affected we discussed this pathway and how it was involved in either AD or PD.

Point # 3.   In the text and table, ‘ptau and pTau” need to be changed as one type either p-Tau or pTau, kindly correct.

Response: Corrected. Thank you for the note!

Point # 4.   In subtopic ‘The role of enormous oxidative stress in triggering Aβ accumulation and p-tau’ (in vitro and in vivo models) is not clearly explained, can add few transgenic mice models which shows  ROS production and oxidative stress in AD in vivo models, please refer https://doi.org/10.3389/fmolb.2022.1050768, https://doi.org/10.3389/fmolb.2022.1030534.

Response: In the subsection entitled:The role of enormous oxidative stress in triggering Aβ accumulation and p-Tau (in vitro and in vivo models)” we involved only data closely related to the topic with an idea to support with available findings on this topic that abnormally elevated oxidative stress might triggers  Aβ deposition and induces pathogenesis of AD.

Point # 5.   TFEB is one of the main modulators of Oxidative stress and improve melatonin functions refer, https://doi .org/10.1038/s41401-021-00711-7.

Response: The key role of TFEB is mentioned in the new version as well as the effect of melatonin on this important to mitophagy signaling molecule in AD.

Point # 6.   The authors need to check the abbreviations, in first time usage they need to use the full word and abbreviation. A separate  section for abbreviation would be better.

Response: We’re grateful for this recommendation of the reviewer and checked carefully when to start usage of abbreviation (after the first mention). A separate section of abbreviation is also included after the Abstract.

Point #7.    The authors need to show the final schematic conclusion diagram revealing a summarized concluding evidence and  mechanism.

Response: The schematic diagram in the conclusion was also prepared and inserted in the new version in Conclusion section.

Point #8.    A careful English check and grammatical errors need to be resolved  by the authors.

Response: We used paid Grammar software again to check and correct if it was necessary all grammatical errors. 

Reviewer 2 Report

The review article “Oxidative stress and aging as risk factors for Alzheimer’s disease and Parkinson's disease: the role of the antioxidant melatonin” by Tchekalarova and Tzoneva is well written. They have successfully presented a comprehensive overview of different factors such as oxidative stress and aging in the development of AD and PD in various model systems. I should appreciate the author’s effort to bring the various pieces of information together. The compilation of various information in form of a table is a good approach. However, in doing so, the manuscript becomes longer. The title of the paper suggests the focus on oxidative and aging as risk factors for AD and PD development and the role of melatonin as an antioxidant system.

A major part of the paper discussed the various model systems for AD and PD rather than talking about the molecular mechanism/pathway affected by changes in redox balance in cells in relation to AD/PD. At the beginning of the review, the authors mentioned the adverse effect of oxidative stress on DNA and RNA but left out various proteins that could also be affected/modified by changes in redox environments and adversely associated with the development or progression of AD and PD.

Oxidative stress is a very general word that could include various types of reactive oxygen species. Each of these ROS can act differently. It will be good to have a more precise discussion about the role of some specific ROS in AD and PD.

Section 2 is very long due to the inclusion of various model systems for PD which is not aligning with the title of the paper. The discussion about different model systems can be shortened and include the study that showed the role of various oxidative stress and aging on PD.

A small section should be included comparing the melatonin vs another antioxidant system (such as the glutathione system) in these diseases. It is not clear why authors focused on melatonin.

Table 2; see row number 4 which is incomplete. There are some grammar and syntax errors/typos present in the manuscript.

Author Response

Reviewer' 2

Point # 1.  Oxidative stress is a very general word that could include various types of reactive oxygen species. Each of these ROS can act differently. It will be good to have a more precise discussion about the role of some specific ROS in AD and PD.  

Response: We’re thankful for this relevant note of the Reviewer. A brief discussion concerning the most common oxidation products was inserted in page 3.

Point # 2.  Section 2 is very long due to the inclusion of various model systems for PD which is not aligning with the title of the paper. The discussion about different model systems can be shortened and include the study that showed the role of various oxidative stress and aging on PD.

Response: We’re thankful to the Reviewer for this relevant message. The correction in this subsections were made accordingly in subsection 2. One of the table was deleted.

Point # 3.  A small section should be included comparing the melatonin vs another antioxidant system (such as the glutathione system) in these diseases. It is not clear why authors focused on melatonin.

Response: In this review the focus of discussion was the role of aging to development of neurodegenerative diseases such as AD and PD.  The concomitant melatonin deficit in aging is an important factor for triggering abnormal oxidative stress and predisposition to Abeta deposition and changed signaling pathways in the above-mentioned diseases. Therefore, the key role of melatonin system and studies with melatonin treatment that supported this hypothesis were cited.

Point # 5.  There are some grammar and syntax errors/typos present in the manuscript.

Response: Corrected. We used paid Grammar software again to check and correct if it was necessary all grammatical errors. 

Reviewer' 2

Point # 1.  Oxidative stress is a very general word that could include various types of reactive oxygen species. Each of these ROS can act differently. It will be good to have a more precise discussion about the role of some specific ROS in AD and PD.  

Response: We’re thankful for this relevant note of the Reviewer. A brief discussion concerning the most common oxidation products was inserted in page 3.

Point # 2.  Section 2 is very long due to the inclusion of various model systems for PD which is not aligning with the title of the paper. The discussion about different model systems can be shortened and include the study that showed the role of various oxidative stress and aging on PD.

Response: We’re thankful to the Reviewer for this relevant message. The correction in this subsections were made accordingly in subsection 2. One of the table was deleted.

Point # 3.  A small section should be included comparing the melatonin vs another antioxidant system (such as the glutathione system) in these diseases. It is not clear why authors focused on melatonin.

Response: In this review the focus of discussion was the role of aging to development of neurodegenerative diseases such as AD and PD.  The concomitant melatonin deficit in aging is an important factor for triggering abnormal oxidative stress and predisposition to Abeta deposition and changed signaling pathways in the above-mentioned diseases. Therefore, the key role of melatonin system and studies with melatonin treatment that supported this hypothesis were cited.

Point # 5.  There are some grammar and syntax errors/typos present in the manuscript.

Response: Corrected. We used paid Grammar software again to check and correct if it was necessary all grammatical errors. 
